# PartialFormer: Modeling Part Instead of Whole for Machine Translation

## Abstract

The parameter redundancy problem in Transformer models has been widely acknowledged in the literature. To address this weakness, we introduce PartialFormer, a parameter-efficient Transformer architecture for machine translation. Compared to previous parameter-efficient Transformer architecture, PartialFormer modifies the modeling strategy of the feed-forward network to allow it to spare tremendous parameters while maintaining large hidden dimension. Additionally, PartialFormer applies two efficient scaling strategies, namely depth scaling and width scaling, to improve performance within a given parameter budget. To efficiently benefit from these scaling strategies, PartialFormer is further enhanced by two cost-effective modifications: 1) a head scaling strategy for efficient width scaling and 2) a residual-like attention calculation for better depth scaling. Extensive experiments on 9 translation tasks validate the effectiveness of our PartialFormer approach.

## 1 Introduction

The Transformer model (Vaswani et al., 2017) has emerged as a cornerstone in the natural language processing (NLP) domain, overshadowing convolutional neural networks (Gehring et al., 2017) and recurrent neural networks (Sutskever et al., 2014) by virtue of its minimal inductive bias, superior scalability, and proficiency in modeling extended sequences. Nonetheless, its substantial computational and parametric requisites pose significant challenges to its deployment and training, warranting an ongoing trend in the research community toward eliminating redundant parameters and computations in the Transformer model (Dehghani et al., 2019; Lan et al., 2020; Reid et al., 2021; Li et al., 2022; Ahmed et al., 2017; Yan et al., 2020; Wu et al., 2020; Mehta et al., 2019, 2021).

Despite their success in improving the parametric and computational efficiency of the Transformer,

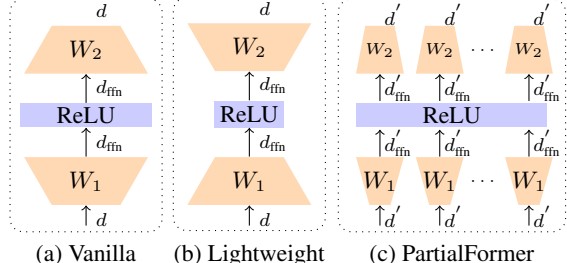

Figure 1: Illustration of our idea.

it is noteworthy that these approaches ignore the importance of feed-forward networks (FFN). Feed-forward networks consume significant parametric and computational overhead due to the inherent large feature space and hidden dimension. To cut down FFNs' overhead, previous studies (Mehta et al., 2021; Wu et al., 2020; Ge et al., 2022) just adopt smaller hidden dimension, e.g., equal to or even lower than the size of feature space. That leads to a question: *Are current lightweight FFNs optimal?*

To address this concern, we turn to the insights provided by Geva et al. (2021), who depicted FFNs as a collection of key-value memories, where the number of memories is equal to the number of hidden dimensions in FFNs. This finding underscores the significance of hidden dimension in FFNs. Drawing inspiration from this finding and the successful application of large hidden sizes in FFNs as evidenced by Meta's 4B model (Tran et al., 2021)[1], we postulate that a truly efficient lightweight FFN should maintain, if not enlarge, the hidden dimension while reducing parameters.

To this end, we propose PartialFormer, an innovative approach to Transformer architecture. The central design of PartialFormer is the Partial-Level Gated Feed-Forward Networks (PG-FFN). We designed the PG-FFN as a set of smaller FFNs in

---

[1]They have shown enlarging the hidden size of FFNs to 16384 delivers significant BLEU improvements.

unison, each producing lower-dimensional hidden features, yet collectively matching or exceeding the hidden dimension of a conventional larger FFN. Moreover, we further equipped PartialFormer with two cost-effective operations: a head scaling strategy for efficient width scaling, and a residual-like attention calculation for stable optimization. These techniques empower PartialFormer to achieve deeper layer stacking or increased width within the same parameter budget.

The strength of PartialFormer has been affirmed through rigorous empirical evaluations on 9 machine translation tasks. Remarkably, even while maintaining similar parameter consumption, our PartialFormer consistently surpasses the vanilla Transformer, employing the same layer depth and embedding width, by an average of 1.29 BLEU points across all 6 WMT'17 machine translations. Furthermore, it achieved a BLEU score of 29.56 on the challenging WMT'14 En-De task with only 68 million parameters, showcasing its effectiveness and efficiency. Our work with PartialFormer thus marks an important step towards the goal of optimized Transformer architectures, marrying performance with efficiency in a manner that has potential for broad impact in NLP applications.

## 2  Preliminary: Transformer

In this section, we present some prior knowledge about the Transformer. Typically, Transformer block always consists of a multi-head self-attention and a feed-forward network. Let $X \in \mathbb{R}^{T \times d}$ be a $T \times d$ input matrix of $T$ tokens. Each multi-head self-attention component owns $H$ heads. For simplicity, we ignore the layer-normalization operation and residual connection.

**Multi-Head Self-Attention** MHSA aims to model the global dependency among tokens. MHSA computes as follows:

$$A^i = \text{Softmax}(\frac{Q^i (K^i)^\mathsf{T}}{\sqrt{d_k}}), \qquad (1)$$

$$\text{head}_i = A^i V^i, \qquad (2)$$

$$X = \sum_{i=1}^{H} \text{head}_i W_i^O, \qquad (3)$$

where $Q^i, K^i, V^i$ denote the query, key and value of $i$-th head, which are derived from input with three learnable matrics $W_i^Q, W_i^K, W_i^V \in \mathbb{R}^{d \times d_k}$ as follows: $Q^i = XW_i^Q, K^i = XW_i^K, V^i =$

$XW_i^V$, respectively. $W_i^O \in \mathbb{R}^{d_k \times d}$ is a learnable matrix. $A^i$ and $\text{head}^i$ denote the attention matrix and representation of $i$-th head, respectively.

**Feed-Forward Network** Feed-forward network is responsible for improving the expressiveness of the whole representation space by adopting an "expansion-activation-reduction" mapping strategy. It computes as follows:

$$X = \text{ReLU}(XW_1 + b_1)W_2 + b_2, \qquad (4)$$

where $W_1 \in \mathbb{R}^{d \times d_{\text{ffn}}}, W_2 \in \mathbb{R}^{d_{\text{ffn}} \times d}, b_1 \in \mathbb{R}^{d_{\text{ffn}}}, b_2 \in \mathbb{R}^d$ as learnable matrices and $d_{\text{ffn}}$ denotes the hidden dimension in FFN that is usually set to $4d$.

## 3  PartialFormer

### 3.1  Overall Architecture

Figure 2 illustrates the overall architecture of PartialFormer, encompassing both an encoder and a decoder. Although the foundational structure adheres to the design of the vanilla Transformer (Vaswani et al., 2017), there are some notable modifications.

**Encoder.** Different from vanilla Transformer, each encoder layer in PartialFormer consists of a unified sub-layer that integrates the PG-FFNs into the multi-head self-attention mechanism rather than separate two sub-layers.

**Decoder.** Each decoder layer is composed of two types of sub-layers, both of which integrate the multi-head attention mechanism with PG-FFNs. The sub-layers differ based on the type of multi-head attention mechanisms employed, specifically whether it's a decoder self-attention or an encoder-decoder cross-attention mechanism.

### 3.2  Information Flow in Unified Sub-Layer

Taking the Encoder as an instance. Each unified sub-layer first computes the multiple attention scores via Eq. (5), then obtains the multiple head features $\{\text{head}^i | 1 \leq i \leq H\}$ via Eq. (2), which is the same as vanilla Transformer. Then, using multiple small FFNs, it processes these head features and ultimately combines the representations via a fusion function according to Eq. (7). That is to say, the PG-FFN is encapsulated into the multihead-attention mechanism.

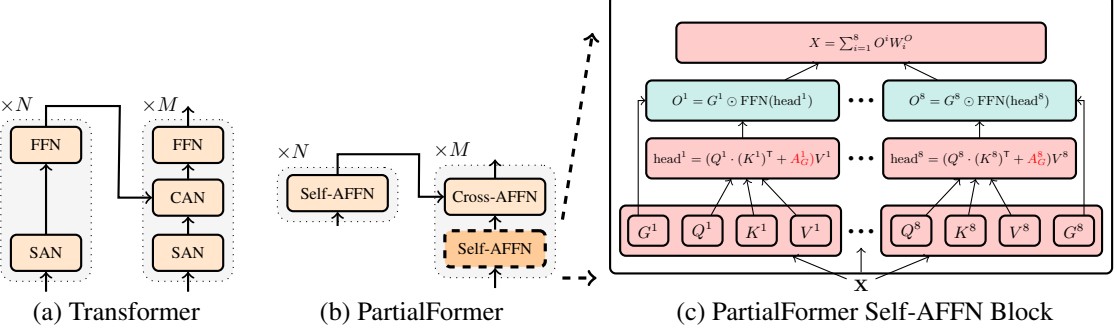

Figure 2: (a) Architecture of Transformer. (b) Architecture of PartialFormer. (c) Details of Self-AFFN Block. All architecture are based on pre-normalization strategy. We omit the layer normalization operation, residual connection, softmax operation and scale coefficient for simplicity.

$$A^i \quad = \quad \text{Softmax}(\frac{Q^i(K^i)^\mathsf{T}}{\sqrt{d_k}} + A^i_G), \quad (5)$$

$$O^i \quad = \quad \text{PG-FFN}(\text{head}^i), \quad (6)$$

$$X \quad = \quad \sum_{i=1}^{H} O^i W^O_i \quad (7)$$

### 3.3 Partial-Level Gated FFN

**Intuition** Previous studies (Wu et al., 2020; Mehta et al., 2021; Ge et al., 2022) have commonly reduced the parameters in feed-forward networks by decreasing the hidden dimension (e.g., 2048 to 256). In contrast, we tackle this issue through a matrix factorization approach. Our key idea involves utilizing a collection of small FFNs to model smaller input features, rather than relying on a single large FFN.

Assume a FFN with mappings of 1024->4096->1024, which consumes around 8.4 million parameters. By decomposing this into 8 smaller FFNs with mappings of 128->512->128, we can retain the same hidden dimension, such as 8 * 512, while using only 1.05 million parameters. This approach significantly reduces parameters while maintaining the crucial desired hidden dimension, as emphasized in previous studies (Geva et al., 2021; Tran et al., 2021).

Furthermore, we have observed that the Transformer architecture inherently consists of multiple smaller subspaces, namely "heads" within the multi-head attention (MHA) mechanism. These heads act as sub-components of the original inputs and retain substantial information from the original data. As a result, PG-FFNs should naturally be constructed based on the MHA mechanism.

**Calculation of PG-FFNs** While group transformation operations could be used to instantiate our idea, they are not optimal on GPUs due to their low I/O efficiency (Ma et al., 2018), causing significant inference latency. To address this, we propose sharing parameters across each FFN within different heads, thereby eliminating the need for group transformation operations.

However, directly sharing weights may result in homogeneous representations across different heads, which may potentially hinder the performance (Li et al., 2018). To mitigate this, we further introduce a head-specific gated mechanism. The core idea is to use a set of diverse masks to filter the information of different heads so that the head representation will be more diverse. Formally, given a set of smaller features $\{\text{head}^i|1 \le i \le H\}$ and diverse masks $\{G^i|1 \le i \le H\}$, the Eq. (6) can rewritten as:

$$O^i \quad = \quad G^i \odot \text{FFN}(\text{head}^i), \quad (8)$$

where $\text{FFN}(\cdot)$ is the same as Eq. (4).

**Generating $\{G^i|1 \le i \le H\}$** In our preliminary experiments, we observed significant diversity in the features generated by different parameters from sub-layer inputs, e.g., $\{V^1, \ldots, V^H\}$. Motivated by this finding, we generate diverse masks in the following manner:

$$G^i = \sigma(X W^G_i), \quad (9)$$

where $W^G_i$ is a learnable matrix and $\sigma$ denotes the activation function, e.g., ReLU, Sigmoid and Tanh. We compare them in Table 8.

### 3.4 Efficient Scaling Strategy

Though PG-FFN offers the advantage of reducing lots of parameters when applied directly to the

transformer, it also leads to performance degradation. Thus, a crucial aspect of this study is to determine how to effectively utilize the spared parameters. In this work, we adopt a hybrid scaling strategy, combining both width scaling and depth scaling, which has been validated in computer vision, e.g., EfficientNet (Tan and Le, 2019).

### 3.4.1 Enabling Efficient Depth Scaling for PartialFormer

Wang et al. (2019); Dong et al. (2021); Wang et al. (2022) have shown that the original location of FFNs plays an essential role in optimizing transformers, e.g., alleviating *Token Uniformity*. Thus, we need to consider the impact brought by the change of FFNs. While the densely residual connection is an efficient way to alleviate it, they are typically either based on feature level (e.g., DLCL (Wang et al., 2019)) or coupled with the network structure (e.g., Realformer (He et al., 2021)).

To this end, we design a new variant of the residual connection integrated into the attention calculation, while also decoupling from the network architecture. Specifically, the calculation of attention maps consists of two parts: 1) $A_G$, the global part, and 2) $A_L$, the local part. The calculation of $A_L$ remains the same as in the vanilla Transformer, while $A_G$ is computed once by using the original embedding as input through Eq. (1). Inspired by He et al. (2021), to efficiently fuse these components, we add them together and apply a Softmax function, as shown in Eq. (5).

In addition to the benefit of efficient depth scaling (See Appendix F), this approach provides remarkable flexibility in combining different attention mechanisms, specifically tailored to address specific conditions. For instance, it allows for the utilization of local attention to calculate $A_G$ when dealing with small datasets (see Appendix D).

### 3.4.2 Head Scaling: An Efficient Width Scaling for PartialFormer

Existing approach to width scaling, which is based on the embedding size, necessitates the simultaneous scaling of both the encoder and decoder for machine translation tasks. This is primarily because researchers commonly employ shared encoder and decoder embedding. However, taking cues from the achievements of depth scaling, it may be more advantageous to adopt a distinct method for scaling width, similar to the approach used for scaling depth. Here we show how PartialFormer has inher-

ent superiority to achieve so.

The width of a Transformer model typically refers to the widest part of the Transformer. In this context, both the vanilla Transformer and previous lightweight Transformer models have widths that are related to the embedding dimension, such as $4d$ or $d$. Therefore, by increasing the embedding size, we can effectively enlarge their width. However, the width definition in PartialFormer is different and can be expressed as $w = H \times d_{\text{ffn}}$, where $w$ denotes the width of model and $d_{\text{ffn}}$ is associated with the head dimension $d_k$. Consequently, we can expand the width by either increasing the number of heads or enlarging $d_k$. A comparison between these approaches is presented in Table 7. Notably, if the head dimension and number of heads are independent of the embedding dimension, PartialFormer allows for easy scaling of width in different ways within the encoder and decoder components.

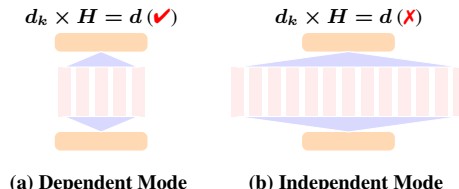

(a) Dependent Mode      (b) Independent Mode

Figure 3: Comparison of ways to generate subspaces in Transformer and PartialFormer.

To this end, we propose a new scaling mechanism, namely head scaling, that scales the width of PartialFormer by directly adding more heads and increasing head dimension, as illustrated in Figure 3. Given the head dimension $d_k$, the embedding dimension $d$, and the number of heads $H$, we consider two strategies to generate $H$ attention heads:

(a) Simple strategy: We employ three learnable matrices, each with a shape of $d \times (d_k \times H)$, to directly obtain the expected number of $Q$, $K$, and $V$.

(b) Complex strategy: we employ a two-step process. First, we generate an intermediate quantity of $Q$ and $K$, and then use a powerful MLP network to expand the attention maps to the desired number. This innovative design draws inspiration from the inherent redundancy found within the attention map (Michel et al., 2019; Clark et al., 2019; Voita et al., 2019), allowing for more heads in PartialFormer under the same parameter budget. We show the comparisons in Table 6.

| Type | Model | N-M | d | $d_k$ | H | MACs | Param | BLEU | COMET-22 |
|---|---|---|---|---|---|---|---|---|---|
| **Multi-Branch Architecture** | Weighted Transformer (Ahmed et al., 2017) | 6-6 | 1024 | - | - | - | 211M | 28.90 | - |
| | Multi-Unit Transformer (Yan et al., 2020) | 6-6 | - | - | - | - | 130M | 29.30 | - |
| | MAT (Fan et al., 2020) | 6-6 | - | - | - | - | 206M | 29.90 | - |
| | Multi-Path Transformer (Lin et al., 2022) | 6-6 | - | - | - | - | 193M | 29.68 | - |
| **Lightweight Architecture** | Evolved Transformer (So et al., 2019) | - | - | - | - | - | 64M | 28.20 | - |
| | Delight (Mehta et al., 2021) | - | 640 | - | - | - | 54M | 28.00 | - |
| **Weight Sharing** | Universal Transformer (Dehghani et al., 2019) | - | 1024 | - | - | - | 65M | 28.90 | - |
| | SubFormer (Reid et al., 2021) | - | - | - | - | - | 63M | 28.50 | - |
| | SubFormer-big (Reid et al., 2021) | - | - | - | - | - | 197M | 29.30 | - |
| | ODE Transformer (RK4) (Li et al., 2022) | 6-6 | 512 | - | - | - | 62M | 29.03 | - |
| | ODE Transformer (RK4) (Li et al., 2022) | 24-6 | 512 | - | - | - | 118M | 29.80 | - |
| **Other Comparisons** | RealFormer (He et al., 2021) | 18-18 | 512 | 64 | 8 | - | 151M | 29.35 | - |
| | DMAN (Fan et al., 2021) | 6-6 | 512 | 64 | 8 | - | 63M | 29.10 | - |
| | Mega-Softmax (Ma et al., 2022) | 6-6 | 512 | - | 1 | - | 67M | 29.01 | - |
| **Our System** | Transformer | 24-6 | 512 | 64 | 8-8 | 11.1B | 118M | 29.05 | 83.60 |
| | PartialFormer (w/o Head Scaling) | 24-6 | 512 | 64 | 8-8 | 8.8B | 66M | 28.86 | 83.35 |
| | PartialFormer | 24-6 | 512 | 64 | 24-16 | 12.2B | 115M | 30.09 | 84.17 |
| | Transformer | 6-6 | 512 | 64 | 8-8 | 9.9B | 62M | 27.43 | 82.19 |
| | Transformer | 24-6 | 360 | 45 | 8-8 | 6.3B | 62M | 28.00 | 82.72 |
| | PartialFormer (w/o Head Scaling) | 24-6 | 360 | 45 | 8-8 | 5.2B | 36M | 27.88 | 82.49 |
| | PartialFormer | 24-6 | 360 | 45 | 24-16 | 6.8B | 61M | 29.23 | 83.74 |
| | PartialFormer | 24-6 | 360 | 45 | 30-16 | 6.9B | 68M | 29.56 | 83.94 |

Table 1: Results on the WMT'14 En-De task. MACs denote the multiplication-addition operations. We compute them via 20 source and target tokens following Mehta et al. (2021).

## 4 Experimental Setups

In our evaluation, we assess the performance of PartialFormer across 9 machine translation tasks[2]. More details are given in Appendix A

**Dataset.** We evaluate our approach on three widely-used datasets: WMT'14 English-German (En-De), WMT'14 English-French (En-Fr), and WMT'16 English-Romanian (En-Ro). Besides, to further validate the effectiveness of PartialFormer, we also evaluate PartialFormer on six translation tasks from WMT'17 benchmark. We preprocess the raw data following the standard strategy.

**Architectures and Selected Baselines.** We use a 24-6 encoder-decoder PartialFormer architecture for its strong performance, on all 9 machine translation tasks. Detailed configurations are provided in the results tables. We compare our approach with various baselines, including vanilla Transformer models, multi-branch architecture, lightweight architecture, weight-sharing methods, and other strong baselines.

**Training & Evaluation.** We train all the models on GeForce RTX 3090 cards via Fairseq (Ott et al., 2019) toolkit. For evaluation, we utilized

multi-BLEU (Papineni et al., 2002) and COMET-22 (Rei et al., 2022) scores. Beam sizes were 4, 4, and 5 for En-De, En-Fr, and En-Ro tasks respectively. *Length_penalty* of 0.6, 0.8, and 1.3 were applied to En-De, En-Fr, and En-Ro tasks respectively. For the WMT'17 benchmark, beam size and *Length_penalty* were set to 4 and 1, respectively. We used an ensemble of the last ten checkpoints.

## 5 Experiments

**Results of WMT'14 En-De** Table 1 presents the results for the WMT'14 En-De task. Note that we also provide a "strong" baseline which also benefits from deep model stacking. Even though the performance of PartialFormer (w/o Head Scaling) is slightly inferior to that of the Transformer model (27.88 vs. 28.00 and 28.86 vs. 29.05), it outshines the latter in terms of parameter efficiency, consuming significantly fewer parameters (36M vs. 62M, 66M vs. 118M). We attribute this phenomenon to our PG-FFN, which leverages a group of compact FFNs. This approach enables PG-FFN to maintain high hidden dimension, while drastically reducing parameter consumption.

Upon utilizing our head scaling technique to amplify the capacity, our Partialformer delivers a BLEU score 29.56 and 30.09 on two configurations, respectively. This surpasses the standard Transformer by 1.56 BLEU points (29.56 vs. 28.00) and

---

[2]We also tested the efficacy of PartialFormer on the language modeling task. Results are shown in Appendix.

| Model | N | d | $d_k$ | H | Param | BLEU |
|---|---|---|---|---|---|---|
| Weighted Transformer (2017) | 6 | - | - | - | 211M | 41.40 |
| Evolved Transformer (2019) | - | - | - | - | 64M | 40.60 |
| Delight (2021) | - | 640 | - | - | 54M | 40.50 |
| ODE Transformer (2022) | 6 | - | - | - | 69M | 42.56 |
| ODE Transformer (2022) | 24 | - | - | - | 123M | 43.28 |
| Multi-Path Transformer (2022) | - | - | - | - | 168M | 42.44 |
| Transformer | 24 | 512 | 64 | 8-8 | 120M | 42.33 |
| PartialFormer (w/o Head Scaling) | 24 | 512 | 64 | 8-8 | 68M | 41.68 |
| PartialFormer | 24 | 512 | 64 | 24-18 | 119M | 43.10 |
| PartialFormer | 24 | 512 | 64 | 24-24 | 127M | 43.29 |
| Transformer | 6 | 512 | 64 | 8-8 | 63M | 40.79 |
| Transformer | 24 | 360 | 45 | 8-8 | 64M | 40.96 |
| PartialFormer (w/o Head Scaling) | 24 | 360 | 45 | 8-8 | 38M | 40.44 |
| PartialFormer | 24 | 360 | 45 | 24-18 | 63M | 42.16 |
| PartialFormer | 24 | 360 | 45 | 24-24 | 67M | 42.39 |

Table 2: Results on the WMT'14 En-Fr task.

| Model | N | d | $d_k$ | H | Param | BLEU |
|---|---|---|---|---|---|---|
| Delight (Mehta et al., 2021) | - | 640 | - | - | 53M | 34.70 |
| Subformer (Reid et al., 2021) | - | - | - | - | 48M | 34.70 |
| ODE Transformer (Li et al., 2022) | 6 | 1024 | 64 | 16-16 | 226M | 35.28 |
| Transformer | 24 | 512 | 64 | 8-8 | 111M | 35.00 |
| PartialFormer (w/o Head Scaling) | 24 | 512 | 64 | 8-8 | 59M | 35.07 |
| PartialFormer | 24 | 320 | 40 | 24-24 | 48M | 35.30 |

Table 3: Results on the WMT'16 En-Ro task.

| Model | Fi⟷En | | De⟷En | | Lv⟷En | | Avg. |
|---|---|---|---|---|---|---|---|
| | Fi→En | En→Fi | De→En | En→De | Lv→En | En→Lv | |
| Transformer | 26.07 | 22.14 | 35.04 | 28.59 | 17.59 | 16.23 | 24.27 |
| PartialFormer | 27.48 | 23.35 | 35.60 | 29.91 | 19.65 | 17.37 | 25.56 |

Table 4: Results on the WMT'17 benchmark. Partial-Former has the same depth and $d$ as the Transformer but consumes 1M fewer parameters on average.

1.04 BLEU points (30.09 vs. 29.05) within a similar model capacity. The enhancement here can be attributed to the head scaling method, which allows PartialFormer to possess a larger hidden dimension, thereby bolstering its capacity for memory storage (Geva et al., 2021). These observations are further confirmed by the COMET-22 scores.

Moreover, PartialFormer can even surpass all selected multi-branch Transformers while using fewer parameters. Notably, PartialFormer ($N = 24, d = 512$) outperforms the latest multi-path Transformer (Lin et al., 2022) by 0.41 BLEU points with 78M fewer parameters. This highlights the efficiency of building a multi-branch network based on inherent subspaces. Additionally, PartialFormer excels over previous lightweight approaches and outperforms state-of-the-art weight-sharing methods, e.g., ODE Transformer (Li et al., 2022), and other strong baselines, e.g., Mega (Ma et al., 2022). Notably, both ODE Transformer and Mega utilize relative position encoding (Shaw et al., 2018).

**Results of WMT'14 En-Fr** Table 2 presents the results of PartialFormer on the WMT'14 En-Fr task. Similar to the findings in the En-De task, PartialFormer demonstrates a similar phenomenon. Notably, PartialFormer achieves comparable results to Transformer ($N = 24, d = 512$) (42.39 vs. 42.33) while utilizing 53M fewer parameters (67M vs. 120M). This highlights the remarkable parameter efficiency of PartialFormer.

**Results of WMT'16 En-Ro** Table 3 presents the results on the test set of the WMT'16 En-Ro task. Notably, PartialFormer achieves the highest BLEU points among all selected baselines. It is particularly remarkable that PartialFormer achieves sim-

ilar results to ODE Transformer while utilizing 178M fewer parameters. This highlights the exceptional efficiency of PartialFormer.

**Results of WMT'17 Benchmark** Table 4 presents the WMT'17 benchmark results, showing that PartialFormer consistently outperforms Transformer by an average of 1.29 BLEU points in all six translation tasks. This finding is consistent with the observed performance in the En-De task.

# 6 Analysis

## 6.1 Ablation Studies

Table 5 presents an ablation study of PartialFormer on the WMT'14 En-De task, demonstrating the critical role of each component. Omitting any element causes performance decline, underscoring the holistic design. The PG-FFN removal (#3 vs. #4) results in a large performance drop of 2.05 BLEU points, despite a mere 16 million parameters reduction. This evidence corroborates previous findings (Dong et al., 2021) on the subpar performance of pure attention networks sans FFN, highlighting the essential role of PG-FFN in PartialFormer.

Besides, Table 5 shows the results of different PartialFormer configurations on the WMT'14 En-De task. The encoder-decoder PartialFormer achieves the highest performance, reaching 29.56 BLEU points, indicating the effectiveness of our approach in enhancing both the encoder and the decoder. Employing our concept to either the encoder or the decoder individually also improves performance, yet the encoder-decoder configuration persistently surpasses others, marking the greatest performance improvement.

| # | Model | Param | BLEU |
|---|---|---|---|
| 1 | Transformer ($N = 24$, $d = 360$) | 62M | 28.00 |
| 2 | Pure Attention ($N = 24$, $d = 360$) | 31M | 25.70 |
| 3 | PartialFormer | 68M | **29.56** |
| 4 | w/o Partial-level Gated FFN | 52M | 27.51 |
| 5 | w/o Residual-like Attention Calculation | 66M | 29.26 |
| 6 | w/o Head Scaling | 36M | 27.88 |
| 7 | PartialFormer (encoder only) | 67M | 29.15 |
| 8 | PartialFormer (decoder only) | 63M | 28.80 |

Table 5: Ablation studies on WMT'14 En-De task.

| Model | Param | BLEU |
|---|---|---|
| PartialFormer (w/o Head Scaling) | 36M | 27.88 |
| + Simple Head Scaling | 68M | 29.33 |
| + Complex Head Scaling | 68M | **29.56** |

Table 6: Comparison of head scaling strategy on WMT'14 En-De task.

| Model | Setting | $H$ | $d$ | $d_k$ | Param | BLEU |
|---|---|---|---|---|---|---|
| PartialFormer | Basic | 30-16 | 360 | 45 | 68M | 29.56 |
| | Varying Encoder $H$ | 24-16 | 360 | 45 | 61M | 29.23 |
| | | 16-16 | 360 | 45 | 51M | 29.02 |
| | Varying Decoder $H$ | 16-24 | 360 | 45 | 56M | 28.85 |
| | | 16-30 | 360 | 45 | 60M | 29.20 |
| | Varying $d^h$ | 30-16 | 360 | 30 | 49M | 28.70 |
| | | 30-16 | 360 | 60 | 86M | 29.68 |
| | | 30-16 | 360 | 90 | 124M | 30.00 |
| | Varying $d$ | 30-16 | 180 | 45 | 35M | 27.61 |
| | | 30-16 | 270 | 45 | 51M | 28.80 |
| | | 30-16 | 450 | 45 | 84M | 29.41 |

Table 7: Comparison of different width scaling strategy on the En-De task.

## 6.2 Comparison of Head Scaling Strategy

Table 6 presents the results of PartialFormer on the En-De task test set with varying head scaling techniques. Both simple and complex strategies effectively utilize additional parameters to enhance PartialFormer's performance. Notably, the complex head scaling technique, allowing for more parameters allocated to additional heads, demonstrates superior performance.

## 6.3 Discussions on Width Scaling Strategies

Table 7 presents the results of analyzing three key ways to increase the width in PartialFormer: 1) $d_k$, 2) $H$, and 3) $d$, on the En-De task's test set. Notably, the findings indicate that both increasing $H$ and adding $d_k$ can effectively enhance the capacity of PartialFormer. Additionally, enlarging $d$ can be beneficial for performance improvements when it is small, e.g., less than 360. However, beyond a certain threshold, further increments of $d$ become redundant and do not lead to performance gains. This aligns with previous studies (Mehta et al., 2021; Baevski and Auli, 2019) highlighting redundant information in the embedding layer.

## 6.4 Comparison of Gating Strategy

Table 8 presents a comparison of various activation functions used in PG-FFN. The results indicate that the default choice, ReLU activation, yields the best performance. One explanation is that the ReLU activation provides hard masks for filtering the information of different heads, compared to other activation functions. Such hard masks can make different heads more diverse.

## 6.5 Efficiency Analysis

Table 9 exhibits the inference efficiency on the test set of En-De task. It is evident that PartialFormer incurs a reasonable increase in inference cost, which remains within acceptable limits.

## 6.6 Analysis on Behaviours of FFN

**Metric.** Following Zhang et al. (2022), we examine FFN behaviors across four aspects: activation neuron count (namely $n_{\text{act.}}$), FFNs' hidden dimension, activation-neuron ratio (activations divided by hidden dimension, namely $R_{\text{act.}}$), and FFN efficiency (activations divided by parameters, namely $\eta_{\text{ffn}}$). Notably, for PartialFormer, the hidden dimension represents the concatenation of hidden dimensions from all smaller FFNs.

**Results.** Figure 4(a-c) exhibits the results on the En-De test set. It is evident that PartialFormer has a lower activation ratio than the vanilla Transformer, as shown in Figure 4(b). This indicates that PG-FFNs based on matrix factorization present lower utilization of the hidden dimension compared to the vanilla FFNs. However, our PG-FFN is parameter consumption friendly, enabling larger hidden layer dimensions with the same parameter budget (e.g., 5400 vs. 1440). Despite lower utilization of hidden dimension, it can still own more activated neurons, as depicted in Figure 4(a). Additionally, our PG-FFN exhibits higher efficiency compared to vanilla FFNs, as shown in Figure 4(c). Multiple small FFNs, like "Swarm Intelligence" (Bonabeau et al., 1999), outperform large FFNs by leveraging the collective strength of weak individuals.

| Model | Param | BLEU |
|---|---|---|
| PG-FFNs | 68M | 29.56 |
| PG-FFNs with Sigmoid activation | 68M | 29.21 |
| PG-FFNs with Tanh activation | 68M | 29.03 |

Table 8: Comparison of activation functions in PG-FFNs.

| Model | Param | Speed (Tok./s) | Memory | BLEU |
|---|---|---|---|---|
| Transformer | 62M | 4325 | 3.0G | 28.00 |
| PartialFormer (w/o head scaling) | 66M | 3634 | 3.2G | 28.86 |
| PartialFormer | 68M | 3023 | 3.3G | **29.56** |

Table 9: Efficiency comparison between Transformer and PartialFormer in inference.

### 6.7 Analysis on Head Diversity

**Metric.** We select the same metric, namely $D_{output}$, as that in Li et al. (2018) to measure the diversity among head features. In this metric, a larger value indicates a higher level of diversity.

**Results.** From Figure 4(d), we can observe that PartialFormer exhibits more diverse head features compared to the vanilla Transformer, even though the vanilla Transformer already demonstrates diverse features. This aligns with previous study (Li et al., 2018), which demonstrates the positive impact of head feature diversity on the Transformer model's performance. Thus, we conclude that the insertion of FFNs into attention mechanism may be a more optimal design.

### 7 Related Work

**Lightweight Transformers** Many methods have been proposed to improve the parameter efficiency of Transformer architecture. The first line is to directly cut down redundant computations and parameters via a more efficient design such as adopting more efficient transformation operations (Mehta et al., 2019, 2021), integrating different but complementary patterns (Wu et al., 2020) and neural architecture search (So et al., 2019). Another research direction for improving parameter efficiency in the Transformer is weight sharing. The popular cross-layer sharing method is utilized by the Universal Transformer (Dehghani et al., 2019). Reid et al. (2021) propose better performance by freeing the first and last encoder layers and widening the intermediate layers. Li et al. (2022) introduce an ordinary differential equation-inspired weight-sharing method for more precise results. Different from these work, our study focus on the design of

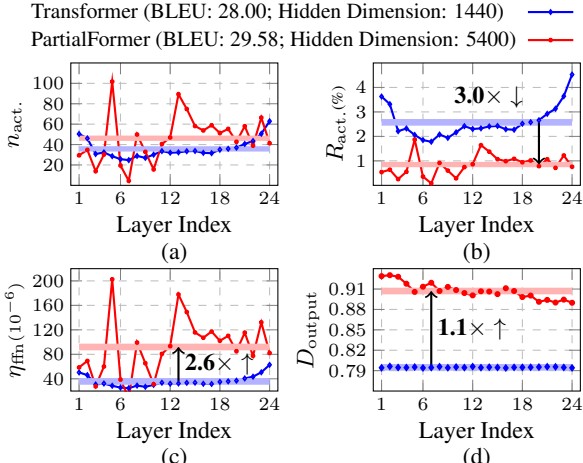

Figure 4: Analysis on behaviours of FFNs and head diversity in Transformer and PartialFormer.

efficient lightweight FFN.

**Multi-Branch Transformer** The multi-branch strategy is widely used in Transformer design. Weighted Transformer (Ahmed et al., 2017) employs a multi-branch FFN, while Multi-attentive Transformer (Fan et al., 2020), Multi-units Transformer (Yan et al., 2020), and Multi-Path Transformer (Lin et al., 2022) extend this concept to different components of the Transformer. Our work introduces a pure multi-branch architecture based on natural subspaces.

**Scaling Strategy in Transformer** Deepening (Bapna et al., 2018; Wang et al., 2019) and widening (Vaswani et al., 2017; Wu et al., 2021) Transformer have been well-acknowledged as two strategies to improve the capacity of Transformer in literature. In this work, PartialFormer adopts two alternative strategies to improve capacity, adding a number of heads and head dimensions.

### 8 Conclusion

In this paper, we present PartialFormer, a new parameter-efficient Transformer architecture that offers an alternative approach to the design of the lightweight FFN. By employing multiple small FFNs and leveraging matrix factorization techniques, PartialFormer effectively reduces the number of parameters in the FFN. Moreover, we propose two innovative operations to further efficiently enhance the model capabilities. Experimental results across various machine translation tasks showcase the significant performance improvements achieved by PartialFormer, while maintaining comparable parameter consumption.

## Limitations

Despite the potential advantages of Partialformer in terms of parameter utilization and performance within a limited parameter budget, it is important to note that the existing conclusions regarding its effectiveness have not been thoroughly examined in the context of large-scale datasets and a higher number of parameters. Further research is needed to validate the claims and assess the scalability of Partialformer in more challenging scenarios.

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

## A  Detailed Setups of Experiments

### A.1  Dataset

Table 10 displays the statistics of all the 9 translation task.

### A.2  Training Details

Table 11 and 12 exhibits the training details on all translation tasks.

## B  Metric Definition

### B.1  Measurement of Head Diversity

Following Li et al. (2018), we measure the head diversity as follows:

$$D_{\text{output}} = \exp(-\frac{1}{H^2} \sum_{i=1}^{H} \sum_{j=1}^{H} \frac{|O^i \cdot O^j|}{\|O^i\| \|O^j\|}) \quad (10)$$

During evaluation, we calculate the metric on all samples and average the values to obtain the final result.

## C  More Comparison with Previous Lightweight Transformer

Table 13 presents a comprehensive comparison of previous lightweight Transformer models on the En-De task's test set, with a specific focus on operating within a smaller parameter budget. The results prominently showcase the outstanding performance of PartialFormer, even when faced with constraints on model capacity. This outcome further emphasizes the superior capabilities of Partial-Former in scenarios with limited resources.

## D  PartialFormer with Different $A_G$ for Small Dataset

Table 14 showcases the results of PartialFormer on the WMT'16 En-Ro task, a small-scale translation dataset, specifically when $A_G$ is calculated using local attention (Shaw et al., 2018). Notably, these results reveal that by adopting such an approach, PartialFormer achieves an impressive BLEU score of 35.76. We hope this can shed lights on the area of model integration.

## E  PartialFormer with GLU and Weight Sharing

In this section, we investigate the integration of PartialFormer with two prominent techniques to enhance parameter efficiency: 1) the weight sharing method (Lan et al., 2020), and 2) gated linear

| Dataset | Sentence | | | BPE | Vocab |
|---------|----------|-----|------|-----|-------|
| | **Train** | **Dev** | **Test** | | |
| WMT'14 En-De | 4.5M | 2999 | 3003 | 32K | 34040 |
| WMT'14 En-Fr | 36M | 26815 | 3003 | 32K | 37288 |
| WMT'16 En-Ro | 0.6M | 1999 | 1999 | 20K | 19064 |
| WMT'17 En-De | 5.9M | 7998 | 3004 | 32K | 35488 |
| WMT'17 De-En | 5.9M | 7998 | 3004 | 32K | 35448 |
| WMT'17 En-Fi | 2.7M | 4225 | 3002 | 32K | 32584 |
| WMT'17 Fi-En | 2.7M | 4225 | 3002 | 32K | 32584 |
| WMT'17 En-Lv | 4.5M | 2003 | 2001 | 20K | 32368 |
| WMT'17 Lv-En | 4.5M | 2003 | 2001 | 20K | 32368 |

Table 10: The details of datasets of 9 translation tasks.

| Hyper-parameter | WMT'14 En-De | WMT'16 En-Ro | WMT'14 En-Fr |
|-----------------|--------------|--------------|--------------|
| GPUs | 8 | 4 | 8 |
| Batch Size | 4096 | 4096 | 4096 |
| Update Frequency | 2 | 1 | 8 |
| Optimer | Adam | Adam | Adam |
| $\mathrm{Adam}_\beta$ | (0.9, 0.997) | (0.9, 0.997) | (0.9, 0.997) |
| LR | 0.0020 | 0.0020 | 0.0020 |
| LR scheduler | inverse sqrt | inverse sqrt | inverse sqrt |
| Initial LR | $1e^{-7}$ | $1e^{-7}$ | $1e^{-7}$ |
| Total updates | 50K | 25K | 100K |
| Warmup updates | 16000 | 8000 | 16000 |
| Weight decay | 0.0000 | 0.0000 | 0.0000 |
| Label smoothing | 0.1 | 0.1 | 0.1 |
| Dropout | 0.1 | 0.1 | 0.1 |
| Attention dropout | 0.1 | 0.1 | 0.1 |
| ReLU dropout | 0.1 | 0.1 | 0.1 |

Table 11: The training setups of WMT'14 En-De, WMT'16 En-Ro and WMT'14 En-Fr tasks.

| Hyper-parameter | En-{De, Lv} | {De, Lv}-En | En-Fi | Fi-En |
|-----------------|-------------|-------------|-------|-------|
| GPUs | 8 | 8 | 8 | 8 |
| Batch Size | 4096 | 4096 | 4096 | 4096 |
| Update Frequency | 2 | 1 | 1 | 4 |
| Optimer | Adam | Adam | Adam | Adam |
| $\mathrm{Adam}_\beta$ | (0.9, 0.997) | (0.9, 0.997) | (0.9, 0.997) | (0.9, 0.997) |
| LR | 0.0020 | 0.0020 | 0.0020 | 0.0020 |
| LR scheduler | inverse sqrt | inverse sqrt | inverse sqrt | inverse sqrt |
| Initial LR | $1e^{-7}$ | $1e^{-7}$ | $1e^{-7}$ | $1e^{-7}$ |
| Total updates | 50K/17K | 50K/17K | 40K | 10K |
| Warmup updates | 16000 | 16000 | 16000 | 16000 |
| Weight decay | 0.0000 | 0.0000 | 0.0000 | 0.0000 |
| Label smoothing | 0.1 | 0.1 | 0.1 | 0.1 |
| Dropout | 0.1 | 0.1 | 0.1 | 0.1 |
| Attention dropout | 0.1 | 0.1 | 0.1 | 0.1 |
| ReLU dropout | 0.1 | 0.1 | 0.1 | 0.1 |

Table 12: The training setups of WMT'17 benchmark.

units (Dauphin et al., 2017). To ensure the utilization of the latest advancements, we employ a state-of-the-art weight sharing method called ODE Transformer (Li et al., 2022), known for its effectiveness in promoting parameter efficiency in Transformer architectures. Additionally, we incorporate Swi-GLU (Shazeer, 2020), a widely adopted GLU-variant that has served as a foundational component in numerous expressive Transformer architectures.

Table 15 displays the results of combining Par-

| Model | Param | BLEU |
|---|---|---|
| DELIGHT (Mehta et al., 2021) | 23M | 26.70 |
| EdgeFormer (Ge et al., 2022) | - | 26.90 |
| Lite Transformer (Wu et al., 2020) | - | 26.50 |
| PartialFormer | 27M | **27.50** |
| Evolved Transformer (So et al., 2019) | 48M | 27.70 |
| DELIGHT (Mehta et al., 2021) | 37M | 27.60 |
| ODE Transformer (Li et al., 2022) | 37M | 28.24 |
| PartialFormer | 36M | **28.35** |

Table 13: Comparison with state-of-the-art models of smaller capacities on the En-De task.

| $A_G$ | $A_L$ | Param | BLEU |
|---|---|---|---|
| RPR | MHSA | 62M | 35.76 |

Table 14: Results of several PartialFormer variants on the En-De task.

| Model | Param | BLEU |
|---|---|---|
| PartialFormer | 67M | 29.56 |
| PartialFormer + Weight Sharing | 67M | 29.71 |
| GLU-based PartialFormer | 67M | 29.67 |

Table 15: Results of PartialFormer variants on the En-De task.

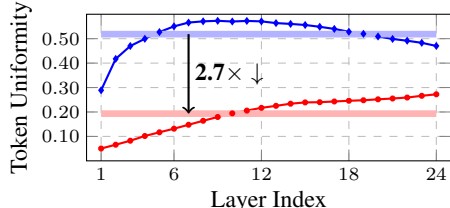

Figure 5: Comparison of token uniformity (lower is better) in Transformer and PartialFormer.

PartialFormer can also show better results compared to strong baseline, e.g., Adaptive Input Transformer (Baevski and Auli, 2019). We will present more comprehensive experiments in the future.

| Model | Depth | $\theta$ (M) | Test PPL |
|---|---|---|---|
| Adaptive Input | 8 | 147M | 21.11 |
| PartialFormer | 16 | 143M | **19.87** |

Table 16: Results on the WikiText-103 dataset.

tialFormer with weight sharing and gated linear units. Despite the integration of these two techniques, the performance gains are marginal. This could be attributed to the fact that PartialFormer already possesses high parameter efficiency, leaving little room for additional enhancements from other technologies. In other words, PartialFormer is inherently a high parameter efficiency architecture.

# F Analysis on Token Uniformity

Following (Dong et al., 2021; Wang et al., 2022), we measure the token uniformity among token representations. We use pearson correlation to compute it.

From Figure 5, we can observe that PartialFormer owns a lower token uniformity among token representations than the vanilla Transformer, revealing that PartialFormer can benefit from depth scaling efficiently (Dong et al., 2021; Wang et al., 2022).

# G Preliminary Experiments on Language Modeling

We also evaluate the effectiveness of PartialFormer on the language modeling task. We can see that

