# OpenReview forum: "PartialFormer: Modeling Part Instead of Whole for Machine Translation"
_EMNLP/2023/Conference — Submitted to EMNLP 2023_

### Official Review · Reviewer_kL8C · 2023-07-19

**Soundness:** 3

**Excitement:**

3: Ambivalent: It has merits (e.g., it reports state-of-the-art results, the idea is nice), but there are key weaknesses (e.g., it describes incremental work), and it can significantly benefit from another round of revision. However, I won't object to accepting it if my co-reviewers champion it.

**Paper Topic And Main Contributions:**

This paper proposes to improve the parameter efficiency of Transformer by merging the FFN layer into each attention head and using a shared low input dimension FFN, and to improve the performance with the saved parameters by modeling gate mechanisms in each attention head, using more attention heads and a global attention mechanism. Experiments on 9 translation tasks show that approach can lead to better performance than the vanilla Transformer with comparable performance.

**Questions For The Authors:**

A, does the Cross-AFFN sub-layer also contains FFN computation?

B, how about the number of parameters and performance if the FFN in each head was not shared?

C, how many improvements can your method obtain if the base setting (6-layer, 512-dimensional embedding) was used?

**Reasons To Accept:**

The paper presents a strategy to reduce the number of parameters of Transformer (by using a small shared FFN in attention heads), and several methods to improve the performance with the saved parameters (modeling gate mechanism, using more attention heads, and a global attention mechanism). Experiments on 9 translation tasks show the effectiveness of the approach.

**Reasons To Reject:**

A, the paper lacks descriptions for the Cross-AFFN sub-layer, if it also computes both cross attention and FFN in each attention heads like Self-AFFN, this increases the decoder depth in terms of sub-layers.

B, using a shared FFN in different attention heads can obviously saving the parameters, and this does not reduce the computation and it is not surprise to improve the performance by using the saved parameters for more attention heads.

C, the performance of the most widely used 6-layer base model of the proposed architecture is not reported, and it is unclear whether the hyper-parameters were optimized for the approach. If so, optimizing the Transformer hyper-parameters given the parameter budget may lead to higher baselines.

thanks for the response, my concerns on C (and also Question C) are well addressed.

**Reproducibility:**

4: Could mostly reproduce the results, but there may be some variation because of sample variance or minor variations in their interpretation of the protocol or method.

**Reviewer Confidence:**

4: Quite sure. I tried to check the important points carefully. It's unlikely, though conceivable, that I missed something that should affect my ratings.

---

> ### Author Rebuttal · Authors · 2023-08-29
>
> Thanks for your constructive comments on our paper. We hope the following responses could address your concerns and re-evaluate the contribution of our paper.
>
> **W1&Q1: the paper lacks descriptions for the Cross-AFFN sub-layer, if it also computes both cross attention and FFN in each attention heads like Self-AFFN, this increases the decoder depth in terms of sub-layers. does the Cross-AFFN sub-layer also contains FFN computation?**
>
> A1: Thank you for highlighting the lack of comprehensive details concerning the Cross-AFFN sub-layer in our manuscript. We appreciate your feedback and would like to address it as follows:
>
> - **Clarification on Cross-AFFN Description**: You are correct that the Cross-AFFN sub-layer incorporates FFN computation. To ensure the computational efficiency, we've set the hidden dimension of PG-FFNs at twice the size of the input, as opposed to the quadrupling (4 times) used in the encoder. We believe this to be an intuitive and computationally efficient configuration. We will ensure that future versions of the paper include a more explicit description of this element.
>
> - **Comparison with Self-AFFN**: Aside from the fundamental distinction between cross-attention and self-attention, Cross-AFFN and Self-AFFN differ in another key aspect: Cross-AFFN omits the A_G components. This design choice stems from the fact that the transformer decoder block consists of two distinct components and accepts heterogeneous inputs. Consequently, we chose not to integrate A_G into Cross-AFFN.
>
> - **Concerning Decoder Depth**: Although the inclusion of Cross-AFFN could ostensibly increase decoder depth in terms of sub-layers, it's important to note two aspects: a) the computational overhead remains relatively low, approximating the cost of 1.5 FFN layer due to a reduced hidden size, and b) the modest increase in decoder depth is justified by the observed performance gains, rendering it negligible.
>
> - **Performance Considerations**: It's worth highlighting that PartialFormer achieved a BLEU score of 29.56 on the WMT'14 En-De task with a mere 68M parameters. This performance serves as a testament to the efficacy of our approach, even with the addition of extra sub-layers. We view this as further evidence that standard Transformer architectures can be optimized to eliminate redundant parameters.
>
> In light of your feedback, we'll consider revising the paper to include a more in-depth breakdown of the Cross-AFFN sub-layer. This will ensure future readers have a clearer understanding of its structure and functionality.
>
> **W2&Q2: using a shared FFN in different attention heads can obviously saving the parameters, and this does not reduce the computation and it is not surprise to improve the performance by using the saved parameters for more attention heads. how about the number of parameters and performance if the FFN in each head was not shared?**
>
> A2: Thank you for raising this insightful question. We're happy to address your concerns.
>
> - **PartialFormer with Unshared FFNs:** In fact, the sharing mechanism is one of the most crucial components in our PartialFormer as when the FFN in each head is not shared, PartialFormer experiences a significant drop in running efficiency, primarily due to the use of group transformation, which is not well supported by Pytorch.  However,  to address your query further, we conducted experiments on the WMT'14 En-De task, and the results are presented below. As illustrated in the table, a variant of PartialFormer that avoids parameter sharing among the FFNs in different heads exhibits further performance improvements, leading to a BLEU score of 30.08, up from 29.56. However, it's important to note that this gain comes at the expense of increased parameter count and computational inefficiency caused by the group-transformation. While, the kernel optimization would be one of our further investigation!
>
> | Method                      | N-M  | d    | d_k  | H     | MACs | Param | Training Speed | BLEU  | COMET-22 |
> | :-------------------------- | ---- | ---- | ---- | ----- | :--: | ----: | :------------: | :---: | :------: |
> | PartialFormer               | 24-6 | 360  | 45   | 30-16 | 6.9B |   68M |       -        | 29.56 |  83.94   |
> | PartialFormer (not sharing) | 24-6 | 360  | 45   | 30-16 | 6.9B |  101M |     0.34x      | 30.08 |  84.19   |
>
> - **Head Scaling naturally fits for PartialFormer:** Head scaling is a scaling strategy that naturally fits the PartialFormer. We also test the effectiveness of head scaling on vanilla Transformer. To keep fairness, we keep the settings the same as that in PartialFormer. We exhibit the results below. We can obtain the following observations:
>   - When we apply head scaling to the vanilla Transformer, it yields an improvement of 0.51 BLEU points, which is considerably less than the 1.68 BLEU points observed with the PartialFormer. It's important to note that a part of this significant performance boost may be due to the larger increase in the number of parameters for the PartialFormer (32M) compared to the vanilla Transformer (21M). More concretely, it is obvious that our ParitialFormer can obtain 0.0525 BLEU gains per million parameters, significantly outperforms that of vanilla Transformer (0.0243). The observation here again indicates that the head-scaling is more suitable for the design of PartialFormer, which is an important contribution of this paper.
>
> | Method                           | N-M  |  d   | d_k  |     H | Param | BLEU  | COMET-22 |
> | -------------------------------- | :--: | :--: | :--: | ----: | ----: | :---: | :------: |
> | Transformer                      | 24-6 | 360  |  45  |   8-8 |   62M | 28.00 |  82.72   |
> | Transformer + Our Head Scaling   | 24-6 | 360  |  45  | 30-16 |   83M | 28.51 |  83.09   |
> | PartialFormer (w/o Head Scaling) | 24-6 | 360  |  45  |   8-8 |   36M | 27.88 |  82.49   |
> | PartialFormer                    | 24-6 | 360  |  45  | 30-16 |   68M | 29.56 |  83.94   |
>
>
>
>
>
> **W3&Q3:  the performance of the most widely used 6-layer base model of the proposed architecture is not reported, and it is unclear whether the hyper-parameters were optimized for the approach. If so, optimizing the Transformer hyper-parameters given the parameter budget may lead to higher baselines. how many improvements can your method obtain if the base setting (6-layer, 512-dimensional embedding) was used?**
>
> A3: Our approach serves as a universal strategy to bolster the Transformer architecture. We conducted experiments using the base configuration, which incorporates a 6-layer structure and a 512-dimensional embedding. The results are presented below. It's evident that, even under this configuration, PartialFormer surpasses the Transformer by a margin of 1.17 BLEU points and 1.02 COMET-22 scores, while maintaining a comparable parameter consumption.
>
> | Method        | N-M  |  d   | d_k  |   H   | Param | BLEU  | COMET-22 |
> | :------------ | :--: | :--: | :--: | :---: | :---: | :---: | :------: |
> | Transformer   | 6-6  | 512  |  64  |  8-8  |  62M  | 27.43 |  82.19   |
> | PartialFormer | 6-6  | 512  |  64  | 24-16 |  63M  | 28.60 |  83.21   |

---

### Official Review · Reviewer_BqHF · 2023-08-01

**Soundness:** 4

**Excitement:**

3: Ambivalent: It has merits (e.g., it reports state-of-the-art results, the idea is nice), but there are key weaknesses (e.g., it describes incremental work), and it can significantly benefit from another round of revision. However, I won't object to accepting it if my co-reviewers champion it.

**Missing References:**

N/A

**Paper Topic And Main Contributions:**

The paragraph discusses the parameter redundancy problem in Transformer models for machine translation and presents a solution called "PartialFormer." PartialFormer is a parameter-efficient Transformer architecture that addresses this issue by modifying the modeling strategy of the feed-forward network to reduce parameter usage while maintaining a large hidden dimension. It also employs two efficient scaling strategies, depth scaling and width scaling, to improve performance within a given parameter budget. To further benefit from these scaling strategies, PartialFormer includes a head scaling strategy for efficient width scaling and a residuallike attention calculation for better depth scaling. The effectiveness of PartialFormer is confirmed through extensive experiments on nine translation tasks.

The proposed method has clear motivation and achieves promising results, and the article is well written.


**Questions For The Authors:**

Q1: In Figure 4, why the head diversity of PartialFormer shows obvious irregular fluctuations in different layers, and is not always at a high level.

Q2: Beam size and some training hyper-parameter settings take different values in different translation directions. Are these setting the best setting for PartialFormer or just follow the setting in the previous work directly?

Q3: Have you verified the effectiveness of PartialFormer on tasks other than translation tasks and WikiText-103 language modeling tasks, such as summarization and other tasks.


**Reasons To Accept:**

-	Compared with the standard Transformer, PartialFormer uses parameters more efficiently and has potential value.
-	Improvements on many translation tasks demonstrate the effectiveness of PartialFormer.
-	The article is well written and easy to follow.


**Reasons To Reject:**

-	The article has no obvious weaknesses to reject, except for some minor improvements.
	-	It is recommended to use sacreBLEU instead of multi-BLEU in the main experiment.
	-	I did not find the performance of PartialFormer in 6 encoder-6 decoder layers setting. This Base setting should be included in experiments.


**Reproducibility:**

3: Could reproduce the results with some difficulty. The settings of parameters are underspecified or subjectively determined; the training/evaluation data are not widely available.

**Reviewer Confidence:**

4: Quite sure. I tried to check the important points carefully. It's unlikely, though conceivable, that I missed something that should affect my ratings.

**Typos Grammar Style And Presentation Improvements:**

-	Paper is well written.
-	Some chapters in the appendices need polishing, for example, the introduction to the experiments and the pointer to the tables are missing in Appendix G.

---

> ### Author Rebuttal · Authors · 2023-08-29
>
> Thanks for your constructive comments on our paper. We hope the following responses could address your concerns and re-evaluate the contribution of our paper.
>
> **W1: It is recommended to use sacreBLEU instead of multi-BLEU in the main experiment.**
>
> A1: Thanks for your constructive  suggestion. Previously, we didn't include this metric because it was absent in many of the methods we compared.  Below, we present the sacreBLEU scores of our methods and two recent work, e.g., Mega and ODE Transformer. We can see that sacreBLEU delivers similar phenomenon to BLEU and COMET-22, the improvement is still stable. We will add Sacrebleu metric for more tasks in the improved version of our paper.
>
> | Method                           |  N-M |    d |  d_k |     H |  MACs | Param | BLEU  | COMET-22 | SacreBLEU |
> | -------------------------------- | ---: | ---: | ---: | ----: | ----: | ----: | ----- | :------: | :-------: |
> | ODE-Transformer (RK4)            |  6-6 |  512 |    - |     - |     - |   62M | 29.03 |    -     |   27.9    |
> | ODE-Transformer (RK4)            | 24-6 |  512 |    - |     - |     - |  118M | 29.80 |    -     |   28.8    |
> | Mega-Softmax                     |  6-6 |  512 |    - |     1 |     - |   67M | 29.01 |    -     |   28.0    |
> | Transformer                      | 24-6 |  512 |   64 |   8-8 | 11.1B |  118M | 29.05 |  83.60   |   27.9    |
> | PartialFormer (w/o Head Scaling) | 24-6 |  512 |   64 |   8-8 |  8.8B |   66M | 28.86 |  83.35   |   27.7    |
> | PartialFormer                    | 24-6 |  512 |   64 | 24-16 | 12.2B |  115M | 30.09 |  84.17   |   29.0    |
> | Transformer                      |  6-6 |  512 |   64 |   8-8 |  9.9B |   62M | 27.43 |  82.19   |   26.4    |
> | Transformer                      | 24-6 |  360 |   45 |   8-8 |  6.3B |   62M | 28.00 |  82.72   |   27.0    |
> | PartialFormer (w/o Head Scaling) | 24-6 |  360 |   45 |   8-8 |  5.2B |   36M | 27.88 |  82.49   |   26.8    |
> | PartialFormer                    | 24-6 |  360 |   45 | 24-16 |  6.8B |   61M | 29.23 |  83.74   |   28.1    |
> | PartialFormer                    | 24-6 |  360 |   45 | 30-16 |  6.9B |   68M | 29.56 |  83.94   |   28.4    |
>
>
>
> **W2: I did not find the performance of PartialFormer in 6 encoder-6 decoder layers setting. This Base setting should be included in experiments.**
>
> A2: Thank you for your valuable feedback. We acknowledge that in our pursuit to ensure a fair comparison, we may have overlooked some aspects. It's worth emphasizing that our PartialFormer is a versatile model capable of performing well even in base configurations. Due to the time constraints imposed by the rebuttal period, we have limited our current results to the WMT'14 English-to-German (En-De) task. Rest assured, we plan to enrich our experimental section with additional results in the revised version of the paper.
>
> | Method        | N-M  |  d   | d_k  |   H   | Param | BLEU  | COMET-22 | SacreBLEU |
> | :------------ | :--: | :--: | :--: | :---: | :---: | :---: | :------: | :-------: |
> | Transformer   | 6-6  | 512  |  64  |  8-8  |  62M  | 27.43 |  82.19   |   26.4    |
> | PartialFormer | 6-6  | 512  |  64  |  8-8  |  42M  | 27.15 |  81.75   |   26.1    |
> | PartialFormer | 6-6  | 512  |  64  | 24-16 |  63M  | 28.60 |  83.21   |   27.5    |
> | PartialFormer | 24-6 | 512  |  64  |  8-8  |  66M  | 28.86 |  83.35   |   27.7    |
>
> As depicted in the table, both the head-scaling (line 3 in Table) and depth-scaling (line 4 in Table) variants of PartialFormer outperform the baseline across all three evaluation metrics, while maintaining comparable parameters. We trust that these results sufficiently address your concerns.
>
> **Q1: In Figure 4, why the head diversity of PartialFormer shows obvious irregular fluctuations in different layers, and is not always at a high level.**
>
> A1: The head diversity of PartialFormer does not show irregular fluctuations in different layers. We guess there are some misunderstandings here:
>
> - Figure 4(d) showcases the head diversity analysis where the row indicates the layer index and the column represents the diversity value—a higher value denotes larger diversity. The diversity metric is calculated in accordance with the methodology from Li et al., 2018. As evident from the figure, PartialFormer exhibits more diverse head representations, corroborating existing research that suggests diversity in head representation is advantageous.
> - Figures 4(a-c) aim to elucidate the behaviors of Feed-Forward Networks (FFNs). Specifically, these figures compare the behaviors of PG-FFNs to traditional FFNs. It is observed that the average activation across different layers for PG-FFNs is higher than that for vanilla Transformers. This suggests that a combination of multiple FFNs may be a preferable architecture choice.
> - On the topic of fluctuations in Figure 4(d), we note that the head diversity values are relatively stable, not showing significant variations. As for activation disparities in other metrics, it is worth noting that neural networks, akin to biological neural systems, may have sections that activate more intensively than others. Given the strong performance of PartialFormer, this variability seems to be a feature, rather than a drawback, of the architecture.
>
> **Q2: Beam size and some training hyper-parameter settings take different values in different translation directions. Are these setting the best setting for PartialFormer or just follow the setting in the previous work directly?**
>
> A2: We predominantly utilized hyper-parameter settings from prior work, without specific tuning tailored for PartialFormer. This approach encompasses variations in beam size and other settings depending on the translation direction. For instance, the beam sizes and length penalties for En-De, En-Fr, and En-Ro are 4/0.6, 4/0.8, and 5/1.3, respectively. These choices align with conventional settings in the literature. Moreover, the training hyper-parameters are adopted from the Deep Transformer model as presented by Wang et al., 2019, and were not specifically optimized for PartialFormer. We posit that dedicated hyper-parameter tuning could potentially unlock even higher performance levels for PartialFormer within a more optimized parameter landscape.
>
> **Q3: Have you verified the effectiveness of PartialFormer on tasks other than translation tasks and WikiText-103 language modeling tasks, such as summarization and other tasks.**
>
> A3. Indeed, we assessed the performance of PartialFormer in the summarization task, as detailed below. It's evident that PartialFormer surpasses the vanilla Transformer while using fewer parameters. In the future, we plan to explore its efficacy across a broader range of tasks.
>
> | Method        | N-M  |  d   | d_k  |   H   | Param | ROUGE-1 | ROUGE-2 | ROUGE-L |
> | :------------ | :--: | :--: | :--: | :---: | :---: | :-----: | :-----: | :-----: |
> | Transformer   | 6-6  | 512  |  64  |  8-8  |  61M  |  41.21  |  18.32  |  37.83  |
> | PartialFormer | 6-6  | 400  |  50  | 24-16 |  37M  |  41.50  |  18.60  |  38.25  |
>
> **Typos Grammar Style And Presentation Improvements: 1. Paper is well written. 2. Some chapters in the appendices need polishing, for example, the introduction to the experiments and the pointer to the tables are missing in Appendix G.**
>
> A: Thank you for highlighting the issues we need to improve in our writing. We genuinely appreciate your feedback. In the future version, we will take action on every point you've raised, and place special emphasis on adding more details in Appendix G. Your insights are invaluable in helping us present our work more clearly. Thanks again for guiding us.

---

### Official Review · Reviewer_1vsa · 2023-08-05

**Soundness:** 3

**Excitement:**

3: Ambivalent: It has merits (e.g., it reports state-of-the-art results, the idea is nice), but there are key weaknesses (e.g., it describes incremental work), and it can significantly benefit from another round of revision. However, I won't object to accepting it if my co-reviewers champion it.

**Paper Topic And Main Contributions:**

This paper addresses parameter and computational efficiency improvements for the Transformer model.
The authors propose a method to decompose the feed-forward network (FFN) inside the Transformer into smaller FFNs based on row number factorization.
While this method reduces the number of parameters, it may lead to performance degradation.
They argue that the performance can be maintained by using a scaling strategy, which is also used in EfficientNet.
Experimental results show that BLEU is improved with the same number of parameters as the vanilla Transformer.

**Questions For The Authors:**

A. Why do you quote the ODE Transformer scores without learnable γ_i?
B. Are all the comparison method scores tried under the same conditions?

**Reasons To Accept:**

The idea of dividing the FFN itself into a set of smaller FFNs based on matrix factorization, rather than reducing the hidden dimension in a single large FFN, is excellent and could be incorporated into various transformer-based models.
A scaling method has also been used to overcome the weaknesses of the proposed method, and the ablation study shows that it works well.
They conducted experiments under various conditions to demonstrate the effectiveness of the proposed method.

**Reasons To Reject:**

I have a question about the experimental results: were the scores quoted from the paper for the experimental results shown in Table 1?
The accuracy of fairseq-based translation results varies depending on the tokenization method and other factors. Therefore, it isn't easy to believe that the results in this table were tested under equal conditions.

Also, it quotes ODE Transformer scores without learnable γ_i, and does not seem to be quoting state-of-the-art scores.
For this reason, it is doubtful that scores other than ODE Transformer are also correctly cited.

**Reproducibility:**

3: Could reproduce the results with some difficulty. The settings of parameters are underspecified or subjectively determined; the training/evaluation data are not widely available.

**Reviewer Confidence:**

3: Pretty sure, but there's a chance I missed something. Although I have a good feel for this area in general, I did not carefully check the paper's details, e.g., the math, experimental design, or novelty.

---

> ### Author Rebuttal · Authors · 2023-08-29
>
> Thanks for your constructive comments on our paper. We hope the following responses could address your concerns and re-evaluate the contribution of our paper.
>
> **W: I have a question about the experimental results: were the scores quoted from the paper for the experimental results shown in Table 1? The accuracy of fairseq-based translation results varies depending on the tokenization method and other factors. Therefore, it isn't easy to believe that the results in this table were tested under equal conditions. Also, it quotes ODE Transformer scores without learnable γ_i, and does not seem to be quoting state-of-the-art scores. For this reason, it is doubtful that scores other than ODE Transformer are also correctly cited.**
>
> - We appreciate your question concerning the experimental results shown in Table 1. The scores for the comparative methods were sourced directly from their original papers to maintain a fair comparison. As displayed in the ODE Transformer (Li et al., 2022), the RK4-block method achieves the best result under the base configuration (about 60M parameters). But for deeper and larger configurations, the RK2-learnable variant yielded superior results. We bring the core results below. Consequently, we selected RK4 as the point of comparison for both Table 1 and Table 2 to maintain consistency across evaluations.
>
>   | Method (En-De)                      | Depth |   BLEU    | SacreBLEU |
>   | ----------------------------------- | :---: | :-------: | :-------: |
>   | ODE Transformer (RK2)               |  6-6  |   28.67   |   27.5    |
>   | ODE Transformer (RK2 learnable γ_i) |  6-6  |   28.89   |   27.7    |
>   | ODE Transformer (RK4)               |  6-6  | **29.03** |   27.9    |
>
>   | Method (En-Fr)                      | Depth |   BLEU    | SacreBLEU |
>   | ----------------------------------- | :---: | :-------: | :-------: |
>   | ODE Transformer (RK2)               |  6-6  |   42.08   |   40.1    |
>   | ODE Transformer (RK2 learnable γ_i) |  6-6  |   42.31   |   40.3    |
>   | ODE Transformer (RK4)               |  6-6  | **42.56** |   40.6    |
>
>
>
> - It is important to note that the original ODE Transformer paper did not provide results using the RK4 method for the English-to-Romanian (En-Ro) translation task. Therefore, we used the highest reported scores from their paper for this specific comparison.
>
> - In Table 15, we also present the results of combining our PartialFormer with the ODE Transformer. For this hybrid model, we opted for the RK2-learnable method as our PartialFormer is designed for deep configurations.
>
> - We believe our methodology for selecting these scores is sound, and that the conclusions drawn are robust. Note that we also evaluate the MACs of the ODE Transformer, ensuring it has the same number of parameters as our PartialFormer for comparison. The result can indicate the superiority of PartialFormer in terms of efficiency. Thank you for bringing up this point, as it gives us the opportunity to clarify our methods.
>
>   | Method (En-De)        | Depth |  MACs  | BLEU  |
>   | --------------------- | ----: | :----: | :---: |
>   | ODE Transformer (RK4) |   6-6 | ~11.1B | 29.03 |
>   | PartialFormer         |  24-6 |  6.8B  | 29.23 |
>
> - Although we are confident in our current selection, we also concur with your perspective. With the increase in parameters, the performance of RK2 learnable becomes more pronounced. We intend to further investigate this in the upcoming versions and provide additional comparisons.
>
> **Q1: Why do you quote the ODE Transformer scores without learnable γ_i?**
>
> A1: The ODE Transformer offers several method variants, such as RK2, RK2 with learnable γ_i, and RK4. Based on the experimental findings, the RK4 version emerges as the most effective for configurations with 62M parameters (base configuration). This variant achieves BLEU scores of 29.03 for En-De and 42.56 for En-Fr tasks. Given its superior performance, we prioritize quoting the results of the RK4 variant and also use it as a benchmark when comparing other configurations for the En-De and En-Fr tasks. And we will add the RK2 with learnable coefficients in our improved version. Thanks for pointing out this!
>
> **Q2: Are all the comparison method scores tried under the same conditions?**
>
> - To ensure a fair and consistent evaluation, we have sourced the scores for comparative methods directly from their respective original papers. We acknowledge the potential for performance variation due to differences in training strategies. Therefore, by referencing scores directly from original sources, we aim to provide an equitable benchmark for comparisons.
> - For our implementation, we adhere to the hyper-parameters as proposed by Wang et al., 2019, which include a learning rate of 2e-3 and a warmup of 16,000 steps, using a pre-norm architecture. Given that most subsequent deep Transformer variants also utilize these hyper-parameters, we believe this offers a fair basis for comparison.
> - It is important to clarify that the baseline standard transformer was trained using an identical strategy as our PartialFormer to ensure a fair comparison. For further details on the specific selection criteria for the ODE Transformer, please refer to Appendix A&A1.
>
> We trust that our detailed explanations address your concerns regarding the experimental setup and comparisons with prior work. Our intention is to conduct a rigorous and comparative fair evaluation, and we believe that these clarifications should dispel any remaining doubts. Should you have any further questions or require additional information, we are open to further discussion.
>
> Wang et al., 2019, Learning deep Transformer models for machine translation. ACL 2019

---

### Meta-Review · Area_Chair_duzy · 2023-09-08

**Recommendation:** 3

**Metareview:**

The paper proposes a more paremeter-efficient version of the Transformer model by factorizing the feedforward layers into multiple heads with a merging/gating mechanism inspired by the multihead attention in the self- and cross-attention layers. The results show better translation quality with the same number of parameters when evaluated on WMT14 en-de, WMT16 en-ro datasets (which are rather outdated) and several language pairs of WMT17 dataset.

The reviews view the paper as technically sound (all scores were 3, during the discussion period, one reviewer incresed their score to 4), but moderately exciting (all scores are 3).

---

### Decision · Program_Chairs · 2023-10-07

**Decision:**

Reject

**Comment:**

The paper proposes a more paremeter-efficient version of the Transformer model by factorizing the feedforward layers into multiple heads with a merging/gating mechanism inspired by the multihead attention in the self- and cross-attention layers. The results show better translation quality with the same number of parameters when evaluated on WMT14 en-de, WMT16 en-ro datasets (which are rather outdated) and several language pairs of WMT17 dataset.

The reviews view the paper as technically sound (all scores were 3, during the discussion period, one reviewer incresed their score to 4), but moderately exciting (all scores are 3).